# Systematic mapping of rRNA 2'-O methylation during frog development and involvement of the methyltransferase Fibrillarin in eye and craniofacial development in *Xenopus laevis*

Jonathan Delhermite[1], Lionel Tafforeau[1,2], Sunny Sharma[1], Virginie Marchand[3], Ludivine Wacheul[1], Ruben Lattuca[4], Simon Desiderio[5], Yuri Motorin[3], Eric Bellefroid[5], Denis L. J. Lafontaine[1]*

1 RNA Molecular Biology, Fonds de la Recherche Scientifique (F.R.S./FNRS), Université libre de Bruxelles (ULB), Biopark campus, Gosselies, Belgium, 2 Cell Biology Lab, Research Institute for Biosciences, Université de Mons (UMONS), Mons, Belgium, 3 Université de Lorraine, CNRS, INSERM, EpiRNA-Seq Core Facility, UMS2008/US40 IBSLor and CNRS, UMR7365 IMoPA, Nancy, France, 4 Erasme campus, Université libre de Bruxelles (ULB), Brussels, Belgium, 5 Developmental Biology, Université libre de Bruxelles (ULB), Biopark campus, Gosselies, Belgium

* denis.lafontaine@ulb.be

**Data Availability Statement:** All RiboMethSeq files are available from the 'European Nucleotide

## Abstract

Ribosomes are essential nanomachines responsible for protein production. Although ribosomes are present in every living cell, ribosome biogenesis dysfunction diseases, called ribosomopathies, impact particular tissues specifically. Here, we evaluate the importance of the box C/D snoRNA-associated ribosomal RNA methyltransferase fibrillarin (Fbl) in the early embryonic development of *Xenopus laevis*. We report that in developing embryos, the neural plate, neural crest cells (NCCs), and NCC derivatives are rich in *fbl* transcripts. Fbl knockdown leads to striking morphological defects affecting the eyes and craniofacial skeleton, due to lack of NCC survival caused by massive p53-dependent apoptosis. Fbl is required for efficient pre-rRNA processing and 18S rRNA production, which explains the early developmental defects. Using RiboMethSeq, we systematically reinvestigated ribosomal RNA 2'-O methylation in *X. laevis*, confirming all 89 previously mapped sites and identifying 15 novel putative positions in 18S and 28S rRNA. Twenty-three positions, including 10 of the new ones, were validated orthogonally by low dNTP primer extension. Bioinformatic screening of the *X. laevis* transcriptome revealed candidate box C/D snoRNAs for all methylated positions. Mapping of 2'-O methylation at six developmental stages in individual embryos indicated a trend towards reduced methylation at specific positions during development. We conclude that fibrillarin knockdown in early *Xenopus* embryos causes reduced production of functional ribosomal subunits, thus impairing NCC formation and migration.

Archive' database (accession number PRJEB42253).

**Funding:** JD and RL were recipients of a FRIA PhD fellowship (F.R.S./FNRS, https://www.frs-fnrs.be/en/). LT was the recipient of a 'Chargé de Recherches' fellowship from the Fonds de la Recherche Scientifique (F.R.S./FNRS). Research in the Lafontaine laboratory is supported by the Belgian Fonds de la Recherche Scientifique (F.R.S./FNRS, PDR grant n˚T.0144.20), the European Joint Programmes on Rare Diseases (EJP RD/JTC2019/PINT-MULTI) 'RiboEurope' and 'DBAGeneCure' (grant n˚R.8015.19 and n˚R.8011.20), the Université libre de Bruxelles (ULB https://www.ulb.be/en/ulb-homepage), the Région Wallonne (SPW EER https://www.wallonie.be/fr/acteurs-et-institutions/wallonie/spw-economie-emploi-recherche/departement-de-la-recherche-et-du-developpement-technologique#) ('RIBOcancer' FSO grant n˚1810070 and POC n˚1880014), the Fonds Jean Brachet, the Internationale Brachet Stiftung, and the Epitran COST action (CA16120 https://epitran.eu/). YM was supported by FRCR EpiARN project from Grand Est Region, France. The funders had no role in study design, data collection and analysis, decision to publish, or preparation of the manuscript.

**Competing interests:** The authors have declared that no competing interests exist.

## Author summary

Ribosomes are essential nanomachines responsible for protein production in all cells. Ribosomopathies are diseases caused by improper ribosome formation due to mutations in ribosomal proteins or ribosome assembly factors. Such diseases primarily affect the brain and blood, and it is unclear how malfunctioning of a process as general as ribosome formation can lead to tissue-specific diseases. Here we have examined how fibrillarin, an enzyme which modifies ribosomal RNA by adding methyl groups at specific sites, affects early embryonic development in the frog *Xenopus laevis*. We have revealed its importance in the maturation of cells forming an embryonic structure called the neural crest. Fibrillarin depletion leads to reduced eye size and abnormal head shape, reminiscent of other conditions such as Treacher Collins syndrome. Molecularly, the observed phenotypes are explainable by increased p53-dependent programmed cell death triggered by inhibition of certain pre-rRNA processing steps. Our systematic investigation of the ribosomal RNA 2'-O methylation repertoire across development has further revealed hypomodification at a late stage of development, which might play a role in late developmental transitions involving differential translation by compositionally different ribosomes.

## Introduction

Ribosomes are essential nanomachines responsible for protein production in every cell. They are sophisticated ribonucleoprotein particles consisting of two subunits of unequal size with specialized functions in translation. The small subunit, 40S in eukaryotes, is responsible for decoding the messenger RNA (mRNA) while the large subunit, 60S, is in charge of joining the amino acids together into polypeptides [1].

Ribosome biogenesis entails numerous reactions, including synthesis of the components (four ribosomal RNAs (rRNAs) and eighty ribosomal proteins in human ribosomes) and their modification, transport, and assembly into functional subunits [2,3]. These reactions are aided by numerous *trans*-acting factors that interact transiently with the maturing ribosomal subunits [4,5]. Although surveillance mechanisms monitor the faithful production of ribosomal subunits, cells can produce ribosomes that differ in composition and possibly also in function [6,7].

An important source of ribosome diversity is ribosomal RNA modification [8]. There are two prevalent types of rRNA modifications: 2'-O methylation, guided by antisense box C/D snoRNAs carrying the methyltransferase fibrillarin, and pseudouridylation, guided by antisense H/ACA snoRNAs bearing the pseudouridine synthetase DKC1 [9]. Each human ribosome contains >100 of each of these two types of modifications on specific residues. These are precisely selected during ribosomal subunit biogenesis by Watson-Crick base-pairing between the snoRNAs and rRNA precursors [10]. In addition, there are a few base modifications, mostly methylation, aminocarboxypropylation, and acetylation [11]. Several rRNA modifications are installed close to important functional sites on the ribosomes, where they are widely assumed to optimize ribosome function [9,11,12]. In zebrafish, inhibition of the production of individual box C/D snoRNAs with interfering antisense oligonucleotides has revealed that they are important for development [13].

Recent work has indicated that in human cancer cells [14–17] and during development in zebrafish [18] and mouse [19], some of the sites susceptible to be 2'-O methylated are not modified on every ribosome. Cells can thus produce a heterogeneous population of ribosomes potentially displaying differential translational properties, i.e. preferentially translating specific

mRNAs [20]. Messenger RNAs liable to be differentially translated include molecules harboring particular *cis*-acting elements such as internal ribosome entry sites [16,21]. Differential translation by compositionally different ribosomes may have important repercussions in normal processes such as cell differentiation and embryogenesis [22,23] and in pathological ones promoting diseases such as cancer and neurodegeneration [21,24,25].

Ribosome biogenesis is a ubiquitous process. Nonetheless, ribosome biogenesis dysfunction impacts more severely particular cells in specific tissues, leading to a diverse set of human syndromes collectively known as ribosomopathies [26,27]. Ribosomopathies are congenital or somatic-tissue-specific diseases resulting from mutations in ribosomal proteins or ribosome biogenesis factors. These lead to a shortage of functional ribosomes and generally to a hypo-proliferation phenotype, often accompanied later in life, because of secondary mutations, by an increased susceptibility to developing cancers [28]. The blood and brain are among the prime targets of ribosomopathies, which then manifest as hematopoiesis defects, microcephally, and intellectual disability.

Ribosomopathy-associated symptoms are often deeply rooted in embryonic development. Ribosome biogenesis dysfunction can profoundly impact the development of specific tissues derived or not from the neural crest cells (NCCs). NCCs are multipotent embryonic migratory cells unique to vertebrates. They can specify towards numerous cell types and tissues, including the teeth, peripheral nervous system and eyes, glia, heart, pigmented cells of the skin, and head skeletal structures. Typically, the craniofacial development aberrations associated with the ribosomopathies Treacher-Collins Syndrome and Diamond-Blackfan Anemia (DBA) are caused by neural crest cell maturation problems [26,27]. The bone marrow failure and anemia observed in Shwachman-Diamond Syndrome (SDS), DBA, and dyskeratosis congenita are caused, rather, by defective maturation of non-neural crest cells [27].

The importance of studying ribosome biogenesis factor mutations in animal models is illustrated by multiple examples, including the intellectual disability (ID) recently found to be associated with mutations in METTL5 [29]. This gene encodes a methyltransferase responsible for depositing an $m^6A$ mark close to the decoding site of the small ribosomal subunit [30]. In cultured human cells, *mettl5* deletion does not cause any obvious cell proliferation defect, but its mutation in the fly leads to severe behavioral problems reminiscent of ID [31]. The zebrafish *Danio rerio* [32–35], the frogs *Xenopus laevis* [36,37] and *Xenopus tropicalis* [38–40], and mouse [41,42] have also proved to be powerful models for approaching ribosomopathies functionally.

Here we have studied the involvement of the box C/D snoRNA-associated methyltransferase fibrillarin in early embryo development in *X. laevis*. We have revealed its importance in small ribosomal subunit production and NCC survival. We have also investigated systematically by RiboMethSeq the complete 2'-O methylation repertoire of rRNAs in developing frog embryos, confirming previously mapped positions, identifying new ones, and revealing that specific sites are strongly hypomodified at late stages of embryogenesis.

## Results

### Expression of *fbl*, *ncl*, and *ubtf* during *X. laevis* development

To investigate the role of the methyltransferase fibrillarin (Fbl) in *Xenopus* development, we first established its spatiotemporal expression during early embryogenesis. For comparison, we mapped the expression of two additional ribosome assembly factors, namely Ubtf, important in rRNA synthesis, and nucleolin (Ncl), a late-acting ribosome assembly factor.

We first analyzed the temporal expression of *fbl*, *ncl*, and *ubtf* and found all three genes to be expressed maternally (stages 2, 4, and 6), and throughout development, notably after the midblastula transition (MBT, stage 8) (S1 Fig).

Next, we analyzed the spatial expression of the selected genes by whole-mount *in situ* hybridization (WISH). We selected five developmental stages corresponding to critical time points during embryogenesis: segmentation (stage 4), gastrulation (stages 10–11), neurulation (stage 18), and early and late organogenesis (stages 25–26, and stage 32). Transcripts of all three genes were detected only at the animal pole of cleaving-stage embryos and at gastrulation (Fig 1A, stage 4 and stages 10–11), suggesting asymmetric distribution of the corresponding maternal transcripts. From neurulation on (stage 18), expression of *fbl*, *ncl*, and *ubtf* was observed in the developing neurectoderm. At the tailbud and early tadpole stages (Fig 1A, stages 25–26, and stage 32, respectively), transcripts were detected in the developing eyes, brain (both the midbrain and the hindbrain), and branchial arches. Expression of *ncl* and *ubtf* was also observed in the pronephros at these later stages.

We further analyzed the expression of *fbl*, the focus of this work, in animal cap explants. Animal cap cells are ectodermal cells forming the roof of the blastocoele. These cells are pluripotent and specify at that stage to form epidermis. We analyzed *fbl* expression by RT-qPCR in animal cap explants derived from embryos injected at their animal pole with either *noggin* mRNA, to induce their differentiation to neural tissue, or a combination of *noggin* and *wnt8* mRNAs, to induce their differentiation to neural crest cells (NCCs) [43]. Expression of *fbl* appeared similar in neuralized and non-induced control caps, but it was higher in induced NCCs (Fig 1B). Fbl is thus likely to play an important role in NCCs. As controls, the expression of *krt12* (encoding the epidermal marker keratin, EpK), *sox3* (a neural marker), and *slug* (an NCCs marker) were determined. As expected, *krt12* was found only in non-induced cells, *sox3* was higher in induced neural caps, and *slug* was detected only in caps induced to form NCCs (Fig 1B).

We wondered whether the abundance of *fbl* transcripts in NCCs is an evolutionarily conserved feature. Previously, the distribution of *fbl* transcripts was studied in the mouse embryo, but the insufficient resolution of the analysis at the time did not allow revealing any specific enrichment in the brain area [44]. To address this, we carefully inspected *fbl* expression in E14.5 mouse embryos by sectioned *in situ* hybridization and found abundant *fbl* transcripts in the retina and the neural-crest-derived mesenchymal cells of the head (Fig 1C). This localization is consistent with that observed in frogs.

In conclusion, *fbl*, *ncl*, and *ubtf* appear to be expressed non-uniformly during *Xenopus* embryonic development, with strikingly stronger expression in neural crest cells and their derivatives. The spatial distribution of *fbl* transcripts is similar in mouse. Ribosomes are produced in every cell, and Fbl is essential to ribosome production ([45–47] and see below). Nonetheless, an elevated *fbl* transcript abundance in particular tissues may be associated with increased Fbl protein synthesis. If Fbl is limiting for ribosome synthesis, this increased production might alter the rate or pathway of pre-rRNA packaging, processing or modification.

## Effects of Fbl, Ncl, and Ubtf depletion on *X. laevis* early development

To assess the involvement of Fbl, Ncl, and Ubtf in early embryogenesis, we performed knockdown experiments with antisense morpholinos (MOs) directed against the translation initiation site in their respective mRNAs. Embryos at the two-cell stage were injected unilaterally at the animal pole with 20 ng MO. Injected embryos were fixed at stage 42 and examined for morphological changes (Fig 2A).

Microinjection of a MO specifically targeting Fbl, Ncl, or Ubtf mRNAs resulted in a similar and more or less pronounced alteration of the eye. Between 59 and 82.5% of the injected embryos displayed a reduced eye size accompanied, in 46–59% of cases, by head asymmetry indicative of craniofacial skeleton alterations (Fig 2A and 2B). The area occupied by the retinal

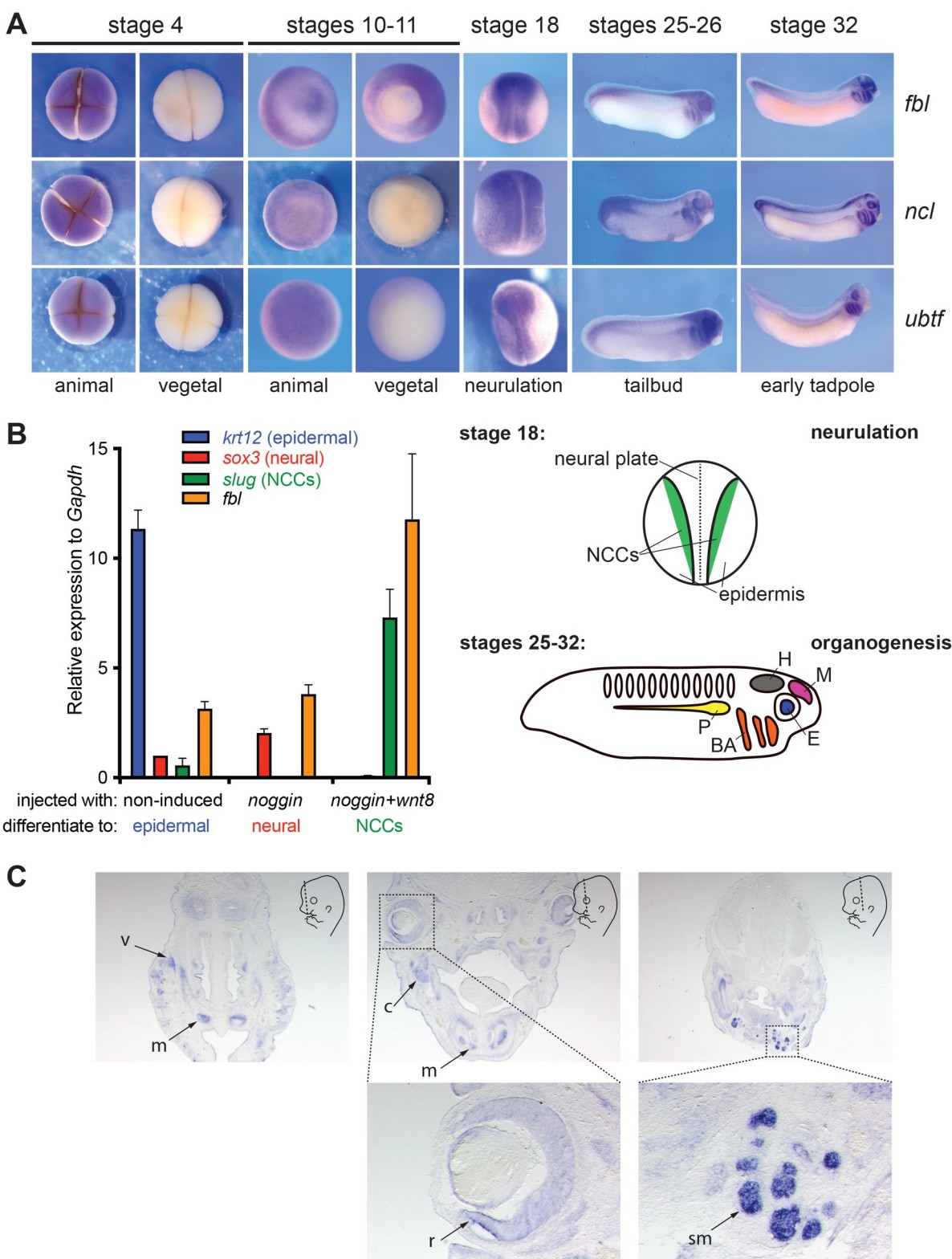

**Fig 1. Spatiotemporal expression of *fbl*, *ncl*, and *ubtf* in *Xenopus laevis* during development. A**, *fbl*, *ncl*, and *ubtf* mRNAs were detected in *X. laevis* during development by whole mount *in situ* hybridization (WISH). The stages and, where relevant, the animal and vegetal poles are indicated. The schematics highlight the labeled structures at stage 18 (neurulation) and at stages 25–32 (organogenesis). NCCs, neural crest cells; H, hindbrain; M, midbrain; E, eyes; P, pronephros; BA, branchial arches. **B**, *fbl* expression was analyzed by RT-qPCR in animal

cap explants injected either with *noggin* (to induce differentiation to neural tissue detected with *sox3*) or with *noggin* and *wnt8* (to stimulate differentiation to neural crest tissue detected with *slug*). Non-induced (non-injected) cap explants naturally differentiate to epiderm detected with *krt12*. As a control, expression of the inspected genes was normalized to *gapdh*. *fbl* was most abundantly expressed in neural crest tissue. **C**, Spatial distribution of *fbl* mRNAs established by WISH in E14.5 mouse embryos. The schematics indicate the orientation of the section plane (dashed line). v, vibrissae; m, mesenchyma; c, primordium of mandibular cartilage; r, retina; sm, submandibular glands.

pigmented epithelium (RPE) was measured and found to be reduced by 10–25% (S2 Fig and S1 Data). No such eye or craniofacial cytoskeleton alterations were observed in embryos injected with a non-targeting control MO (Ctrl). The morphology of the craniofacial cartilage was further investigated by alcian blue staining (Fig 2C). This further revealed in the morphants a striking reduction, or near-complete loss in the case of Fbl depletion, of the branchial arches (Fig 2C, arrowheads). The amount of residual endogenous Fbl after MO treatment was established by western blotting as ~31% (~one-third) of the normal level (Fig 2D, lane 2).

To prove that the observed developmental defects were caused by Fbl depletion, we performed rescue assays. When the *fbl* MO was co-injected with an *in vitro*-synthesized mRNA encoding Flag-tagged wild-type Fbl (Flag-Fbl-wt), the eye size, the area occupied by the retinal pigmented epithelium, the presence of the branchial arches, and the symmetry of craniofacial skeleton were all largely restored (Fig 2B, 2E and S2 Fig and S1 Data). Note that the rescue transcript was made resistant to MO interference by modifying its 5' untranslated region (see Materials and Methods). Efficient expression of Flag-tagged-fibrillarin was confirmed by western blotting (Fig 2D, lane 3). In rescued Fbl morphants, about half-sized normal branchial arches were formed (Fig 2E, green arrowheads). This highly distinctive phenotype was never observed in embryos treated only with *fbl* MO. Such partial morphological rescue is compatible with the amount of Fbl detected in rescued morphants: counting both the endogenous residual protein and the Flag-tagged construct, this amounted to ~49% of the normal level (Fig 2D).

At this stage, it was interesting to probe if the methyltransferase function of Fbl is important for morphological rescue. Considering that the residual amount of endogenous Fbl in the morphants is about one-third of the normal amount, testing this possibility was always going to be challenging. Nonetheless, as an initial approach, we expressed the *fbl*-D238A mutation instead of the wild-type allele in our rescue assay. In the atomic resolution structure of the *Archaeoglobus fulgidus* fibrillarin-Nop5 complex bound to its cofactor and methyl donor *S*-adenosyl-L-methionine (AdoMet), it has been shown that Asp-133 (equivalent to *Xenopus laevis* D238) is situated within 3.5 Å of the thiomethyl carbon of the bound AdoMet. This implies that it plays a role as a catalytic residue [48] (see S3 Fig for the position of the residue in the catalytic pocket of the human protein). When this residue was mutated into an alanine, the methylation activity of fibrillarin was indeed totally abolished in an *in vitro* methylation assay [49]. Interestingly, expression of Flag-Fbl-D238A in morphants rescued the phenotypes (eye size, craniofacial skeleton symmetry, and branchial arches) to the same extent as did Flag-Fbl-wt. In particular, half-sized branchial arches were formed (Fig 2B and 2E). This indicates that providing maturing embryos with an additional supply of Fbl (in this case ~10%, see Fig 2D, lane 4), even if it is not catalytically active, was sufficient to restore development partially.

We conclude that all three tested ribosome biogenesis factors, Fbl, Ncl and Ubtf, are important for eye and craniofacial skeleton development in *Xenopus*.

## Fibrillarin depletion reduces expression of neural border and neural crest markers

To understand the origin of the eye and craniofacial skeleton defects observed upon Fbl depletion, we analyzed the distribution of a series of prototypic developmental markers. Embryos

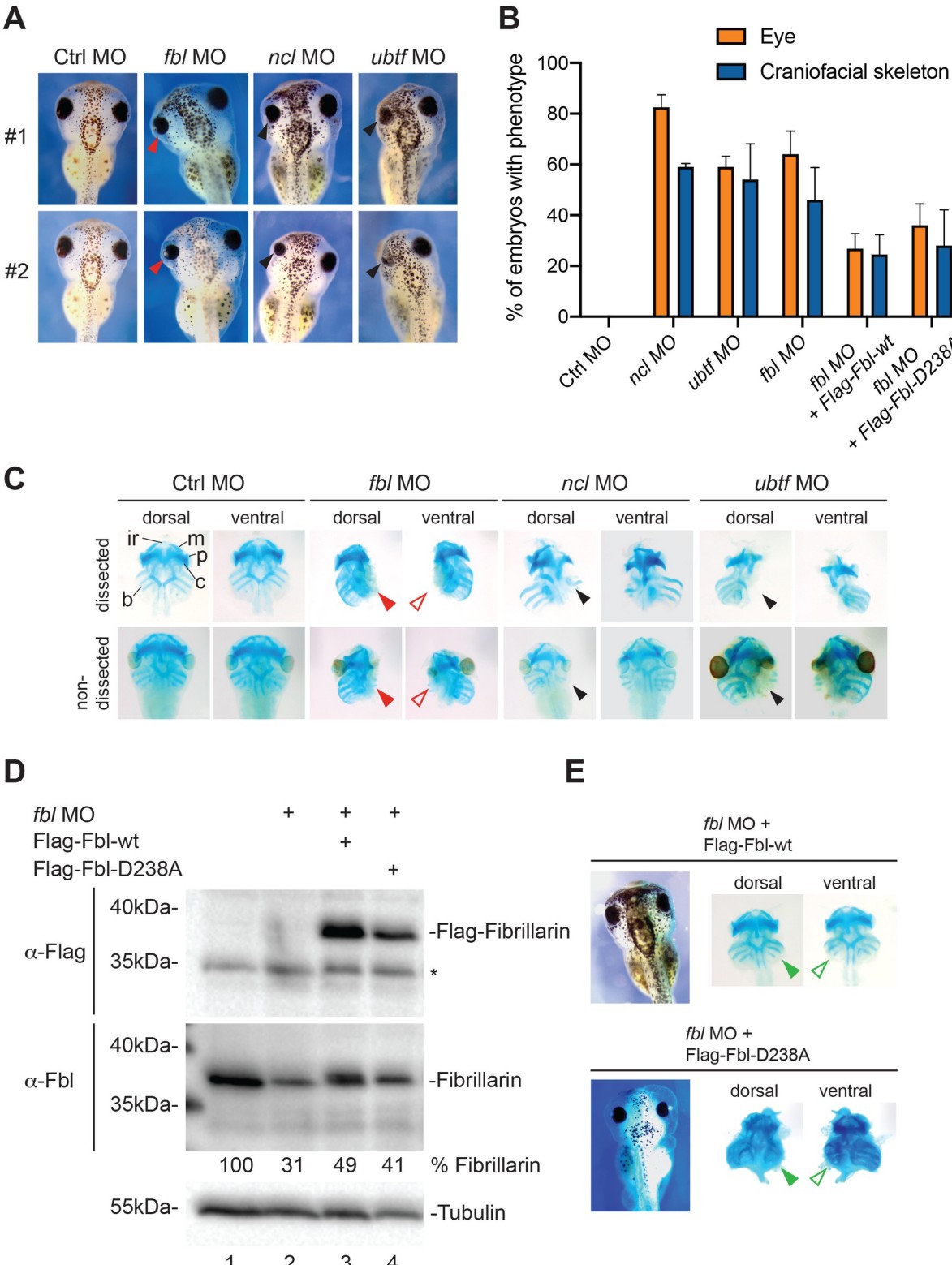

**Fig 2. Morphological defects during *Xenopus laevis* development upon Fbl, Ncl, or Ubtf depletion. A**, Embryos injected unilaterally at the two-cell stage with the indicated morpholino (MO) were inspected under the microscope at stage 42. As a control, a non-targeting morpholino (Ctrl) was used. Two embryos are presented (#1 and #2), with the left side injected. With the Ctrl morpholino and those targeting *ubtf*, and *ncl* expression, two independent experiments were performed, the total number of embryos tested being 100, 94 and 77,

respectively. With the morpholino targeting *fbl* expression, a total of 154 embryos were analyzed in three independent experiments. **B**, Quantification of the data presented in panels A and E. Percentage of embryos displaying reduced eye size, or craniofacial skeleton defects (head asymmetry). **C**, Cartilage staining with alcian blue revealed a defect in all the branchial cartilages, namely: ir, infrarostral; m, meckel; p, palatoquadrate; c, ceratohyal; b, other branchial cartilages. Non-dissected and dissected embryos are presented for comparison. Arrowheads highlight striking reduction, or near-complete loss of the branchial arches. **D**, Western blot analysis of Fbl-depleted and rescued morphants. Total protein was extracted from individual neurula embryos; one embryo equivalent was loaded per lane. The blots were probed with an anti-Flag antibody (top panel) or an anti-Fbl antibody (bottom panel). The anti-Flag antibody detects only the rescue constructs (Flag-Fbl-wt or Flag-Fbl-D238A). The asterisk (*) denotes detection of an aspecific band. The anti-Fbl antibody detects both the endogenous protein and the rescue constructs. The percentage of Fbl expressed with respect to the uninjected control is indicated. The western blots displayed are representative of four independent experiments. As loading control, the blots were probed for alpha-tubulin. **E**, Both the wild-type Flag-Fbl and the catalytically deficient Flag-Fbl-D238A restored formation of branchial arches to about half the normal size. Representative example of an embryo co-injected with *fbl* MO and either a MO-resistant wild-type rescue construct (top) or the same construct harboring the D238A mutation (bottom). A total of 77 and 65 embryos were analyzed in two independent experiments with the Flag-Fbl-wt and Flag-Fbl-D238A construct, respectively.

were injected unilaterally at the two-cell stage with 20 ng MO targeting *fbl* transcripts. The injected embryos were fixed at the neurula stage (stage 15–18) or at organogenesis (early tadpole stage 32) and analyzed by WISH to reveal expression of selected markers (Fig 3A and 3B).

Expression of the neural plate border genes *msx1* and *hairy2* and of the neural crest specifier gene *twist* was reduced on the injected side of the embryos (Fig 3A). This was observed at both the neurula and early tadpole stages for *hairy2* and *twist*, and at the neurula stage for *msx1* (Fig 3A). In contrast, expression of *krt12* (an epidermal marker), *sox2* (a pan-neural plate marker), *pax2* (forebrain), *pax6* (eye), and also *pax8* (otic placodes and pronephros) and *prdm12* (cranial placodes) all appeared unaffected at the neurula stage. Of these, only *pax6* was downregulated in later-stage embryos (stage 32).

Importantly, the downregulation of *twist*, observed in neurula embryos, and that of *pax6*, observed in the eye in tadpole embryos, were both rescued by co-injection of the MO-resistant wild-type *fbl* construct (Fig 3B). Co-injection of the *fbl-D238A* construct also restored proper expression of *twist* and *pax6* at these stages (Fig 3B).

In order to provide initial mechanistic insights into the cellular basis of the phenotypes observed in frog upon fibrillarin knockdown, we assessed apoptotic DNA fragmentation by performing terminal deoxynucleotidyl transferase dUTP nick end labeling (TUNEL) assays (Fig 3C). The *fbl* MO was injected unilaterally at the two-cell stage and TUNEL was applied to whole-mount embryos once they reached the neurula stage. TUNEL revealed important levels of DNA fragmentation in the injected sides of the embryos, particularly obvious at the neural plate, where *fbl* is normally highly expressed. This indicates massive apoptosis. Interestingly, the increased TUNEL staining was suppressed when the *fbl* MO was co-injected with an anti-p53 MO. This is consistent with the role of p53 in promoting apoptosis. As a control, embryos were unilaterally injected with a transcript expressing DMRT5, whose overexpression has been shown to promote apoptosis under these conditions [50].

Thus, Fbl depletion affects neural crest survival as a result of massive p53-dependent apoptosis. Altogether, our results also indicated that the head and eye defects of Fbl morphants are a likely consequence of altered cranial neural crest cell development, as these cells start to migrate into and become patterned within the facial primordia at the neurula stage, coinciding with the observed increased apoptosis.

## Fibrillarin is required for efficient pre-rRNA processing in *X. laevis*

The involvement of fibrillarin in pre-rRNA processing has been established in a variety of eukaryotic models, ranging from budding yeast to human cells. In these models, the protein was shown to be important for the initial pre-rRNA cleavages leading to synthesis of 18S rRNA, the RNA component of the small ribosomal subunit [45–47]. These cleavages largely

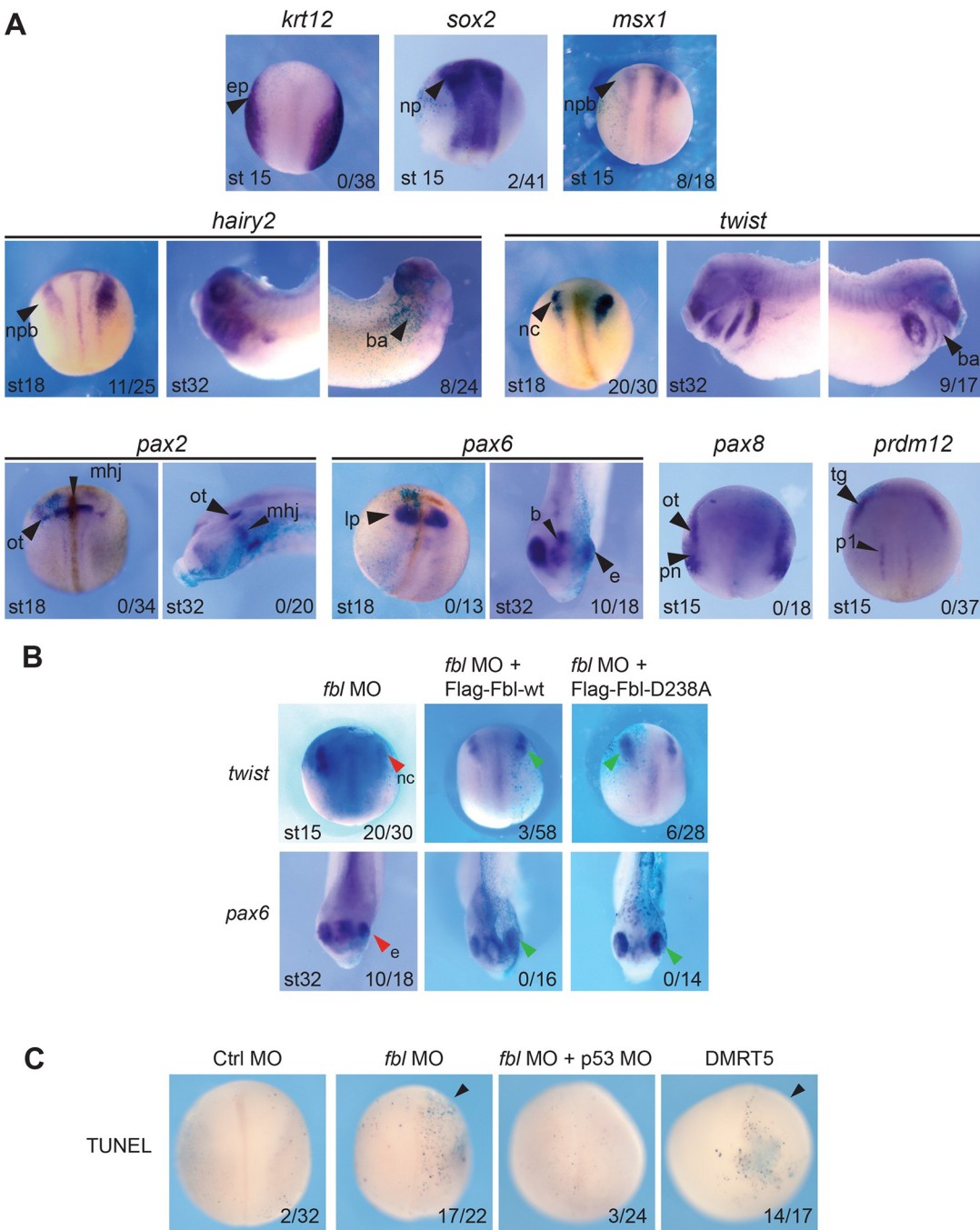

**Fig 3. The distribution of developmental markers is affected upon Fbl depletion in *Xenopus laevis*. A,** Expression patterns of a series of representative developmental markers established by WISH in embryos asymmetrically depleted of Fbl. The embryos were observed at the neurula stage (stages 15–18) or at organogenesis (early tadpole, stage 32). ep, epidermis; np, neural plate; npb, neural plate border; ba, branchial arches; nc, neural crest; ot, otic placode; mhj, midbrain-hindbrain junction; lp, lens placode; b, brain; e, eye; pn, pronephros; tg, trigeminal placode; p1, V1 interneuron progenitors of the spinal cord. The injected side of each embryo was identified by co-injecting *lacZ* mRNA. **B,** The altered distribution of *twist* and *pax6* transcripts, observed upon Fbl depletion, was restored by co-injecting embryos with MO and an mRNAs encoding the wild-type protein (Flag-Fbl-wt) or the D238A (Flag-Fbl-D238A) mutant. Legend as in panel A. The injected side of the embryos was identified by co-injecting *lacZ* mRNA. In panels A and B, the penetrance of the phenotype (number of embryos with reduced expression of the marker *vs.* the total number of observed embryos) is indicated at the bottom right of each panel. **C,** Detection of apoptotic DNA fragmentation by TUNEL. Embryos were unilaterally injected with *fbl* MO alone or together with p53 MO. As controls, embryos were injected with a Ctrl MO or with a transcript encoding DMRT5, known to promote apoptosis. The number of embryos with increased apoptosis *vs.* the total number of observed embryos is indicated.

rely on the box C/D snoRNA U3, conserved in *Xenopus* and shown in this organism to be important for these processing steps [51,52].

To test whether Fbl is involved in ribosome biogenesis in *Xenopus*, total RNA extracted from MO-treated embryos was analyzed by northern blotting at the early tadpole stage (S4 Fig). Probing of the membrane with specific radioactively-labeled oligonucleotides revealed several phenotypes upon Fbl depletion: 1) accumulation of the primary ribosomal transcript (40S), 2) reduction of 20S pre-rRNA, a precursor of the 18S rRNA, and 3) reduced production of mature 18S rRNA (S4 Fig). These phenotypes are consistent with inhibition of early pre-rRNA cleavages and confirm an evolutionarily conserved role of Fbl in small ribosomal sub-unit biogenesis in *Xenopus*.

During the early steps of ribosomal subunit assembly, fibrillarin binds to nascent pre-rRNA transcripts. This early binding step must occur for subunit biogenesis to proceed faithfully (RNA processing) and, importantly, it is independent of its catalytic activity in methylation ([53]). In agreement with our catalytic mutant rescue analysis (Fig 2E), we conclude it is the involvement in pre-rRNA processing of fibrillarin which is responsible for the early develop-mental defects reported. Furthermore, ribosome biogenesis dysfunctions, including pre-rRNA processing inhibitions, are well-established as triggering nucleolar surveillance leading to p53 stabilization (discussed in [54]). The p53-dependent apoptosis observed by TUNEL (Fig 3C) is thus explainable by the involvement of Fbl in pre-rRNA processing.

## Systematic mapping of 2'-O-methylated ribose in *X. laevis* ribosomal RNAs

Fibrillarin modifies pre-rRNAs, aided by box C/D snoRNAs acting as antisense guides [55]. In pioneering work, Prof. Maden in Liverpool, U.K. mapped the sites of 2'-O methylation on *X. laevis* ribosomal RNAs, using RNase finger-printing by 2-D thin-layer chromatography [56–58]. Although quite laborious, relying on the use of heavily radioactively-labeled RNAs, and offering only limited resolution, this approach enabled this investigator, nonetheless, to map 89 of the 98 2'-O methylation sites predicted at the time [56–58]. It remained unclear, however, where on the ribosome the nine remaining sites might lie, whether additional sites exist (the sensitivity of the technique was good but some sites might have escaped detection), and most importantly, what proportion of ribosomes in a population of particles is modified at each sites.

At this stage, although we had established it is the role of fibrillarin in pre-rRNA processing which is important for early development, we nonetheless felt it was interesting to reassess sys-tematically in a quantitative fashion sites of ribosomal RNA 2'-O methylation. Such analysis might reveal undocumented variation at specific positions, possibly at particular stages of development.

To investigate systematically the 2'-O methylation of rRNAs in *X. laevis*, we applied Ribo-MethSeq, a deep sequencing-based mapping technique relying on differential sensitivity of 2'-O methylated nucleotides to mild alkaline hydrolysis [59,60]. RiboMethSeq does not only afford mapping rRNA sugar methylation with single nucleotide resolution, it is also particu-larly powerful for quantitative assessment [14–16]. Specifically, we wanted to establish a com-prehensive repertoire of all sites of 2'-O methylation in *X. laevis* rRNAs (possibly identifying novel sites), to evaluate quantitatively the level of modification at each modified position (pos-sibly identifying sites of hypomodification), and to test the possibility that individual sites might be differentially modified during development.

We selected for analysis six stages across early development: segmentation (stage 2), gastru-lation (stage 10), neurulation (stage 16), early organogenesis (tailbud, stage 23 and tadpole, stage 32), and late organogenesis (stage 45). As RiboMethSeq requires only minute amounts of

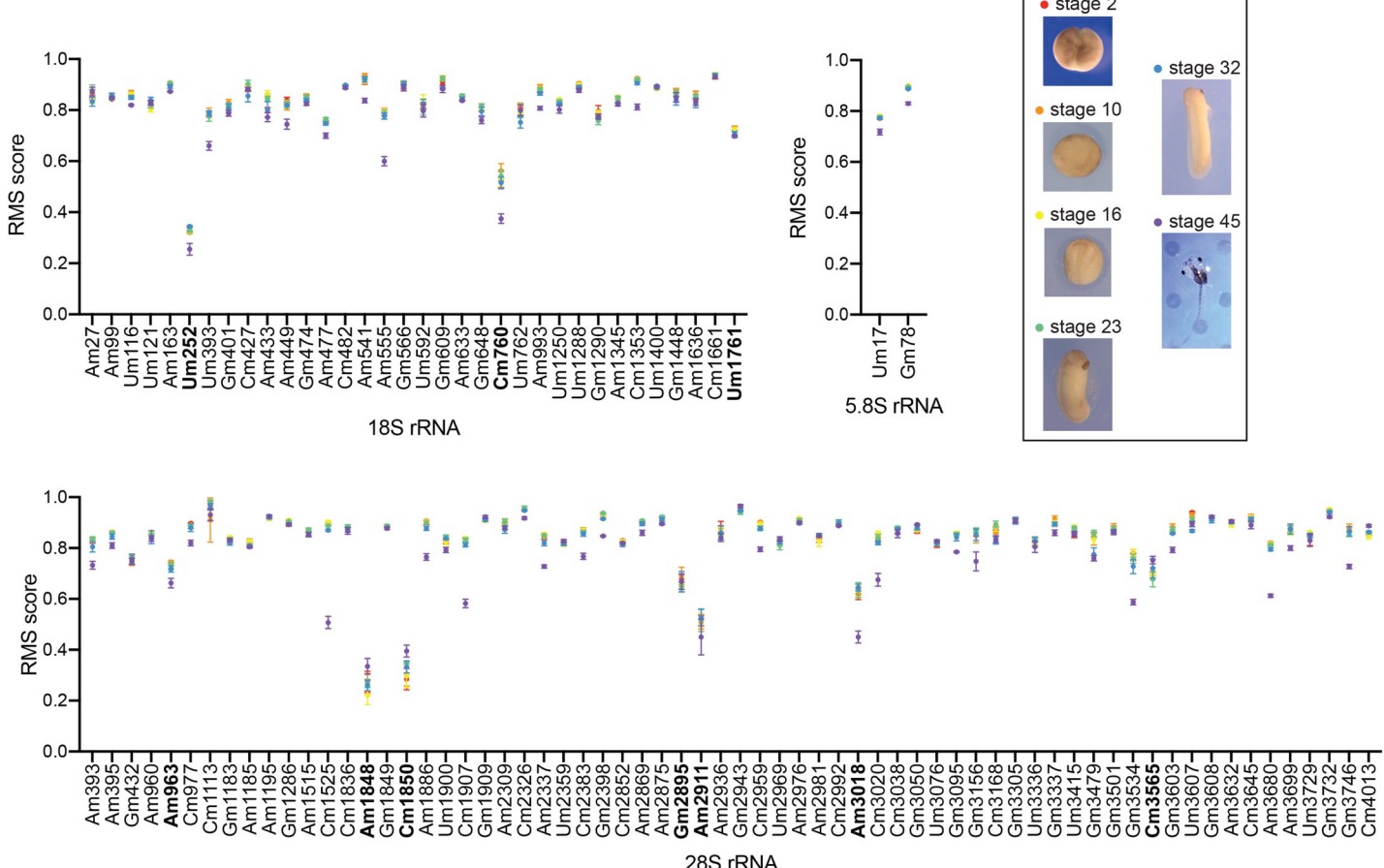

**Fig 4. RiboMethSeq analysis of *Xenopus* 18S, 5.8S, and 28S rRNAs.** The methylation status of rRNAs was investigated systematically by RiboMethSeq at six developmental stages (stages 2, 10, 16, 23, 32, and 45). At each stage, RNA extracted from four individual embryos was sequenced independently. X axis, modified nucleotides. Y axis, methylation scores (see S1 and S2 Tables for values). In bold, positions hypomodified at all stages (using a cut-off of 0.75). Inset, representative pictures of embryos at each analyzed stage are displayed.

total RNA [60], for each stage analyzed we used RNA extracted from a single embryo. For consistency, four embryos were analyzed individually at each selected stage (Figs 4 and S5). Briefly, total RNA was extracted, fragmented by means of mild alkaline treatment, and converted to a deep sequencing library. Libraries were loaded onto a HiSeq1000 sequencer. Following alkaline fragmentation, RNA fragments starting or ending with a 2'-O methylated residue are strongly underrepresented because the phosphodiester bond adjacent to the modification is resistant to cleavage. Mapping of the millions of reads (15 to 25 million) onto the reference rRNA sequences made it possible to infer which position was methylated and the extent of modification at each site (methylation score) [60]. Note that for 28S rRNA, we observed several sequence differences between the reference *X. laevis* sequence in the database and the sequence of our animals. We therefore deposited a revised 28S sequence at NCBI under ref. MT122800. For the other rRNAs, the reference sequences and ours were the same.

The results for 18S, 5.8S, and 28S rRNAs are displayed in Fig 4. The complete dataset is available in S1 and S2 Tables. Cluster analysis revealed, for each of the six developmental stages analyzed, clear clustering of the methylation scores obtained for the four individual embryos (S5 Fig), revealing that our dataset is extremely robust.

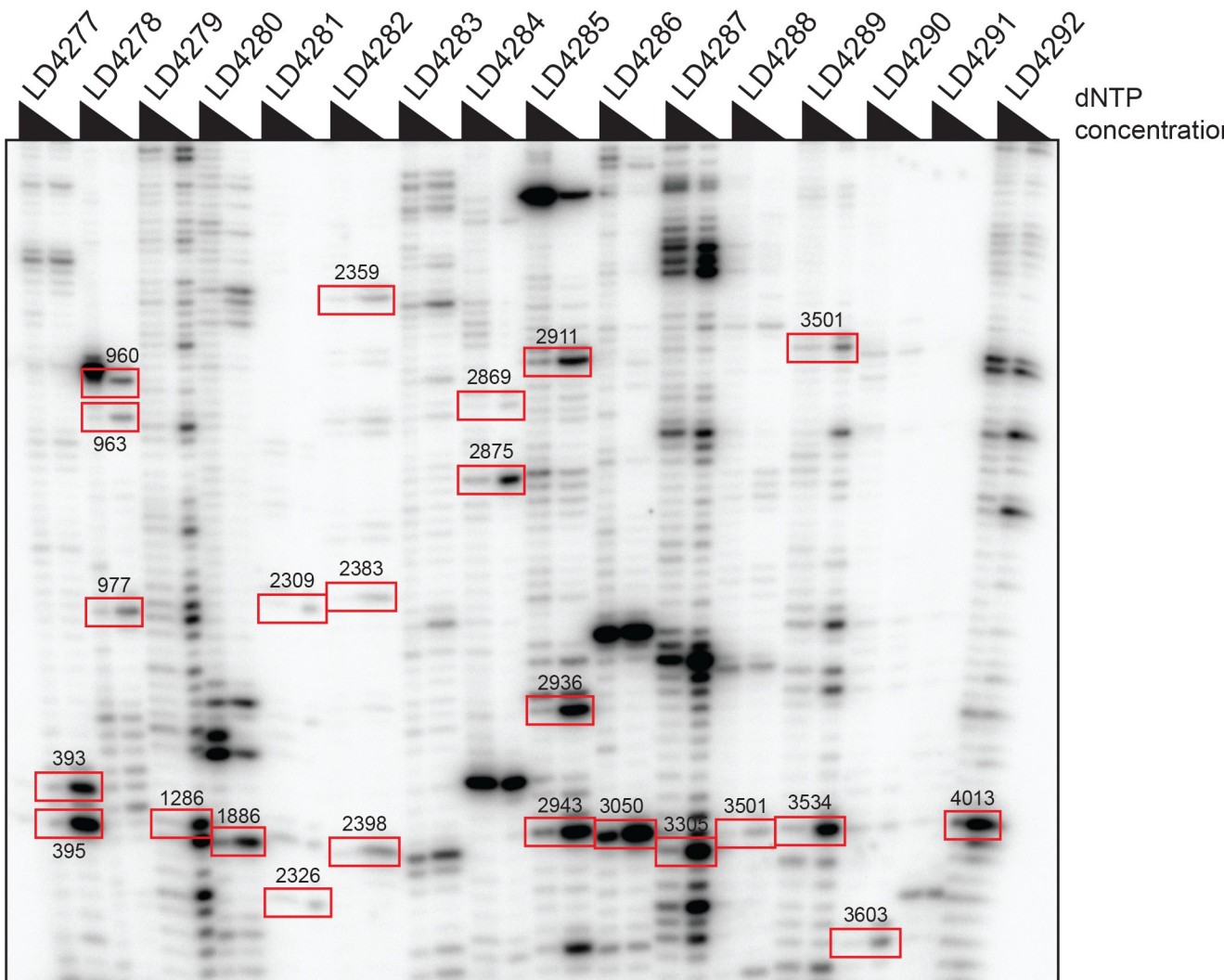

**Fig 5. Orthogonal validation by low dNTP primer extension analysis of newly identified 2'-O methylated positions on *Xenopus* 28S rRNA.** Low dNTP primer extension has proved efficient for specifically detecting 2'-O methylated nucleotides [61]. Total RNA extracted from stage 23 *X. laevis* embryos was analyzed by primer extension using regular or low dNTP conditions with sixteen oligonucleotides (LD4277 to LD4292) designed to survey 23 positions on the 28S rRNA. A position was confirmed when the signal obtained with low-concentration dNTP was higher than that observed with the regular concentration. The analysis confirmed all 23 positions inspected (highlighted by red squares). Note that position 3501 was confirmed twice, once with oligonucleotide LD4288 and once with LD4289.

Several conclusions were drawn from our RiboMethSeq analysis:

Firstly, the analysis confirmed all 89 of the previously mapped 2'-O methylated positions.

Secondly, we identified 15 novel putative 2'-O methylated residues: one on 18S and fourteen on 28S rRNA (Fig 4 and S1 and S2 Tables). We confirmed ten of these new positions, as well as thirteen of the previously mapped ones, by use of an orthogonal mapping technique, namely: low dNTP primer extension detection [61] (Fig 5). The orthogonally-confirmed novel positions are the 28S nucleotides Am393, Am395, Am960, Gm1286, Am1886, Gm2398, Gm2943, Gm3050, Gm3305, and Gm3501 (Fig 5 and S1 and S2 Tables). The previously mapped positions which we confirmed orthogonally are the 28S residues Am963, Cm977, Am2309, Cm2326, Um2359, Cm2383, Am2869, Am2875, Am2911, Am2936, Gm3534, Gm3603, and Cm4013.

A bioinformatic search of the *X. laevis* transcriptome, assisted by the Snoscan software, allowed us to assign box C/D snoRNA guides to all the 2'-O methylated positions identified by

RiboMethSeq (S3 Table). We identified a total of 49 novel *X. laevis* box C/D snoRNA candidates (S3 Table).

Thirdly, using as cut-off a methylation score of 0.75, we observed that numerous positions are not fully modified (Fig 4 and S1 and S2 Tables). This indicates that in *X. laevis* as in other investigated experimental models such as budding yeast, zebrafish, mouse, and human, cells produce heterogeneous populations of ribosomes with differentially modified ribosomes that might display specific translational properties (see Discussion).

Lastly, close inspection of the methylation scores at individual positions through development revealed that specific nucleotides are differentially modified as the embryos develop (Fig 4 and S1 and S2 Tables), with a trend towards reduced methylation levels at specific positions. At stage 45 (highlighted in purple in Fig 4), this was notably the case at one position in 5.8S, three positions in 18S, and eight in 28S.

Of particular interest is the 18S rRNA Um252 modification, a newly identified position in *X. laevis* which is not modified in human cells and which is hypomodified at all the stages inspected. Other positions of interest on 18S rRNA are: (1) Cm760, which is hypomethylated throughout development (interestingly, the corresponding position is also hypomethylated in cultured human cells [15]); (2) Um1761, which is hypomodified throughout development whereas its human counterpart is fully methylated [15]; (3) three additional positions which are hypomethylated only at stage 45 (Um393, Am477, Am555). The equivalent human positions are also hypomodified, at least in some cell lines investigated [14,15,17]. On the 28S rRNA, we noted that: (1) seven positions are hypomodified at all developmental stages analyzed (Am963, Am1848, Cm1850, Gm2895, Am2911, Am3018, Cm3565), the equivalent human positions of five of these also being hypomodified [14,15]; (2) eight positions which are hypomethylated only at stage 45 (Am393, Cm1525, Cm1907, Am2337, Cm3020, Gm3534, Am3680, Gm3746). In the human cell lines tested, the equivalent positions of three of these (Cm1881, Cm3869, and Am4571) are also hypomethylated [15,17].

## Discussion

### Neural crest cell survival requires efficient ribosome biogenesis in *Xenopus*

With the emergence of ribosomopathies, a novel class of human syndromes caused by ribosome biogenesis dysfunction, investigators have become aware of the importance of understanding how tissue-specific diseases can involve a process as global as ribosome biogenesis [26]. In this work, we have studied the involvement of the box C/D snoRNA-associated methyltransferase fibrillarin in the early development of *X. laevis*. We have established the spatiotemporal expression of fibrillarin, revealing it to be abundantly expressed in neural crest cells and cells derived from them (Fig 1). This is also true of two other factors involved in distinct ribosome biogenesis steps, Ncl and Ubtf. We have shown that morpholino-mediated depletion of Fbl strongly impacts the survival of NCCs, with dramatic repercussions on eye and craniofacial skeleton formation leading to head dysmorphism caused by severe branchial arches alterations (Fig 2). We have corroborated these developmental defects by demonstrating altered expression of several markers of NCC formation (*hairy2*, *msx1*, and *twist*) and by detecting increased apoptosis at the neural plate (Fig 3).

### Why are ribosome biogenesis factors more abundantly expressed in neural crest cells and derivatives?

A straightforward explanation might be that specific parts of the embryo rely on more abundant protein production to develop, and thus on more active ribosome biogenesis. Another

possibility is that specific parts of the embryo have differential requirements for specific ribosome biogenesis factors, possibly to assemble distinct ribosomes. Although this second hypothesis is quite appealing, we do not particularly favor it at this stage, because the spatio-temporal expression profiles of the three ribosome biogenesis factors inspected in this work largely overlap and because these are core factors playing evolutionarily conserved essential roles in ribosome biogenesis. This view is strengthened by the observation that additional ribosome assembly factors, and also several ribosomal proteins, have been shown to share, during early development, expression patterns largely similar to that observed here for Fbl, Ncl, and Ubtf [36–39,62].

During development in the mouse, Fbl is essential at a very early stage, prior to embryo implantation [44]. The mechanisms involved and the precise spatiotemporal expression of Fbl in mouse embryos remained undefined. Here we have clarified the pattern of expression of Fbl at high resolution in E14.5 mouse embryos, finding it to be consistent with our observations on *X. laevis* (Fig 1C). In zebrafish, fibrillarin is preferentially expressed in the proliferating cells of the optic tectum, in the retina, and in the dorsal midbrain, and it is important for the maturation of these regions [47]. These observations on a fish are perfectly consistent with our own findings in *X. laevis* (Figs 1 and 3). The function of fibrillarin in pre-rRNA processing is conserved in yeast [45], human cells [46], and zebrafish [47], and now we show it to be conserved also in *X. laevis* (S4 Fig).

There are currently two models for explaining how a global process such as ribosome biogenesis can lead to tissue-specific disorders: the 'specialized ribosomes' model and the 'ribosome concentration' model. The specialized ribosome model posits that compositionally distinct ribosomes, such as those harboring more or less methylated residues, have differential translational capacities [63]. The ribosome concentration model proposes, rather, that the amount of functional ribosomes is what dictates which mRNAs are preferentially translated. In this case, the rationale is that naturally 'less favored' mRNAs will be even less translated when the intracellular ribosome concentration drops [64]. Both of these models have qualities and both mechanisms are likely to coexist in cells, each contributing to some extent to regulating the translational capacity. Because of the pre-rRNA processing defects observed and the phenotypic rescue with a catalytically defective allele, in the case of the Fbl-mediated developmental defects reported here, our data clearly favors the ribosome concentration model.

The majority of bones, cartilages, and connective tissues comprising the head and face are derived from neural crest cells. Craniofacial skeleton alterations are conspicuous in developing animals where ribosome biogenesis is inhibited. Typically, they are observed in fish and mouse embryos with phenotypes reminiscent of Treacher-Collins syndrome, a neurocristopathy caused by mutations in factors important for rRNA synthesis [42,65,66]. They are also observed upon fibrillarin depletion in *Xenopus*, as reported here, and in both *Xenopus* and fish models upon depletion of several other factors intervening at different steps of ribosome biogenesis [33,38]. Altogether, this strongly supports the notion that NCC survival is particularly sensitive to the capacity of cells to synthetize ribosomes.

## Ribosomal RNAs are distinctly hypomethylated at late stages of *Xenopus* development

On the basis of our RiboMethSeq data, we report 103 methylation sites on *X. laevis* rRNA: 34 on 18S, 2 on 5.8S, and 67 on 28S (see S1 and S2 Tables). By comparison, human rRNAs harbor 112 positions: 42 on 18S, 2 on 5.8S, and 68 on 28S, most sites being conserved from frog to human (S1 Table). On 18S rRNA, *X. laevis* carries the Um252 modification, which is not conserved in human cells and which is hypomodified at all stages inspected. Human 18S rRNA

contains nine modifications which are not conserved in *X. laevis*, namely Am159, Um172, Cm174, Um354, Cm621, Gm867, Cm1272, Gm1447, and Um1668. Among them, most are hypomodified in the human cell lines tested (Cm174, Um354, Cm621, Gm867, Cm1272, Gm1447 and Um1668). On 28S rRNA, *X. laevis* harbors four modifications which are not conserved in human cells: Gm432, Cm1113, Gm3534 and Cm4013. Only Cm3534 is hypomethylated at later developmental stages (32 and 45). Human 28S rRNA contains five modifications that are absent in *X. laevis*: Gm1316, Um1773, Cm2824, Gm3627, and Gm4618. Again, most of these are hypomodified (Gm1316, Um1773, Cm2824, Gm4618).

Of the 103 positions mapped here in *X. laevis*, fifteen have not been detected previously [56–58]. Importantly, using of a low dNTP primer extension assay, we have orthogonally validated 23 positions, including ten of the newly identified ones (Fig 5). Furthermore, upon searching the *X. laevis* transcriptome, we have identified box C/D snoRNA candidates for all the identified positions (S3 Table). We also note that one position identified historically [56,57], Um2326, was not confirmed by our RMS or our low dNTP primer extension analysis. This position is also absent in *X. tropicalis*. We conclude that this position was originally misassigned.

We have profiled rRNA sugar methylation in individual embryos at six developmental stages, concluding that the vast majority (87.6%) of positions are robustly installed, while a minority (12.4%) are hypomethylated (cut-off score: 0.75). In principle, fractional ribosomal RNA 2'-O methylation implies that cells produce a heterogeneous population of ribosomes. Whether these differentially modified ribosomes are all actively involved in translation remains to be established. Ribosomal RNA hypomethylation has been observed in cancer cells [14–17], and during development in fish [18] and mouse [19]. We now report it in *Xenopus*. The exact significance of substoichiometric rRNA methylation remains largely unknown.

We have also observed that the degree of hypomethylation increases substantially at later stages of development. This is particularly striking at stage 45, at three positions on 18S, one on 5.8S, and eight on 28S. Whether and how reduced methylation at these sites might affect translation, and possibly late stage of development, deserves future investigation.

The mechanisms of hypomethylation are not yet known. In particular, it remains unclear whether sites of hypomethylation are hardcoded or whether they simply reflect stochastic elements inherent in ribosomal subunit biogenesis. What we have learned so far with cancer cells is that not all sites vary to the same extent [15]. In particular, sites located at the periphery of a ribosomal subunit are more prone to variation than those located at the core of the ribosome, where the functionally important sites lie. Thus, only a limited number of positions are subjected to partial modification.

We have also learned that upon controlled and progressive depletion of fibrillarin in human cells, the sites which become less modified are largely those which are naturally hypomodified [15]. This clearly indicates that some sites are more easily modified than others. It also provides a simple mechanistic explanation of hypomethylation: under conditions of active cell proliferation, as in cancer cells or at specific developmental stages, the rRNA modification machinery might be 'overwhelmed' by increased ribosome production, and snoRNPs might be insufficient in number or might find it hard to access their binding sites within the complex folding 3-D structure of the maturing ribosomal subunits. In agreement with this hypothesis, profiling of rRNA 2'-O methylation in differentiated tissues, where cell proliferation is reduced as compared to cultured cells, shows most sites to be robustly methylated [18,19].

In a recent work, the team of Prof. Gall used RNA-sequencing data from the giant oocyte nucleus of *X. tropicalis* to annotate snoRNAs and to compare his findings with those pertaining to other vertebrates [67]. In their analysis, the authors identified in *X. tropicalis* 102 2'-O methylated positions and 99 box C/D snoRNAs. They predicted novel modified positions,

some of which were investigated in a complex orthologous system consisting of a yeast strain deleted of the endogenous snoRNA guide and provided with an exogenous frog guide expressed from a plasmid. Methylation of the predicted substrate was then verified by low dNTP primer extension [67], which is non- quantitative. While snoRNP machineries are well conserved, the use of a heterologous system might not be optimal in all cases because of species specificities. It also raises questions regarding the copy number of the complementing exogenous snoRNA, particularly in cases of weak anti-sense guide elements which might not allow modification when expressed at physiological levels. By comparison, we have conducted a comprehensive survey of rRNA 2'-O methylation directly on *X. laevis* rRNAs extracted from individual developing animals. For this we have used the highly quantitative RiboMethSeq technique, subsequently assigning snoRNA guides to the known and newly identified positions.

In conclusion, we have shown in *X. laevis* that neural crest cells are particularly sensitive to ribosome biogenesis inhibition, and that in the late stages of embryonic development rRNA hypomethylation strikingly increases at specific positions. Further study is required to determine whether this developmentally regulated rRNA hypomethylation results in differential translation of particular mRNA transcripts important for controlling late developmental transitions.

## Materials and methods

### Ethics statement

Animal housing and experimental protocols were approved by the CEBEA ("Comité d'éthique et du bien être animal") of the IBMM-ULB and conformed to the European guidelines on the ethical care and use of animals.

### Expression plasmids and morpholinos

The *fbl*, *ubtf*, and *ncl* expression vector were constructed by PCR from *X. laevis* pCMV-SPORT6-FBL (mRNAID: XM_018228388.1), pCMV-SPORT6-UBTF (mRNAID: NM_001101787.1), and pCR4-TOPO-NCL (mRNAID: NM_001088088.1), using primers described in S4 Table. The PCR products were cloned into the *BamH* I and *Xho* I sites of the pCS2 vector. To deplete Ubtf, fibrillarin, and nucleolin, we used an antisense morpholino specific to sequences directly upstream of their ATG. Morpholinos were purchased from Gene-Tools (Philomath, OR) (S5 Table).

Fibrillarin rescue constructs: We produced two fibrillarin rescue constructs, one encoding the wild-type allele, the other encoding the D238A mutation (see S3 Fig). mRNA was produced from each construct by *in vitro* transcription, it was capped, poly-adenylated, and purified on gel. The purified mRNA was injected directly into the embryos (see below). The rescue constructs were resistant to the morpholinos because the sequence upstream of the ATG was modified so as to lose its complementarity to the antisense oligo. To distinguish endogenously expressed fibrillarin from protein produced from the injected mRNAs, a Flag-tag was inserted in-frame at the amino-terminus of the rescue constructs. The wild-type construct was amplified with primers LD3036 and LD2749 and cloned into pCS2-Flag between the *EcoR* I and the *Xho* I site. The D238A catalytically deficient *fbl* mutant was generated by site-directed mutagenesis with the help of the QuikChange II site-directed mutagenesis kit (Stratagene) and primers LD3881 and LD3882. Oligonucleotides are listed in S4 Table.

### Embryos, micro-injections, and *X. laevis* animal cap explant culture

*Xenopus* embryos were obtained from adult frogs by hormone-induced egg-laying. Embryos were generated by *in vitro* fertilization [68] and staged according to [69]. For *in situ* analysis,

embryos were injected in one blastomere at the 2–4 cell stage (200 pg mRNA or 10 ng of morpholino per blastomere) and fixed at the neurula or tailbud stages. *X. laevis wnt8* (50 pg/blastomere [70]), *noggin* (100 pg/blastomere [71]), *Flag-fbl-wt* (250 pg/blastomere), and *Flag-fbl-D238A* (250 pg/blastomere) mRNAs were synthesized with the mMessage mMachine kit (Ambion). To reveal the injected side, embryos were co-injected with *lacZ* mRNA (50 pg/blastomere).

In the animal cap assays, mRNAs were microinjected into the animal region of each blastomere of four-cell-stage embryos. Animal caps were dissected at the blastula stage (stage 9) and cultured in 1x Steinberg medium, 0.1% BSA until sibling embryos reached the neurula stage. All *Xenopus* embryo injections were performed on at least three batches of embryos.

The area occupied by the RPE was calculated for both eyes with Image J and the injected to non-injected side area ratio was calculated. RPE ratios were analyzed with the Kolmogorov-Smirnov test (an unpaired t-test, unparamatretric and specific for cumulative distribution comparisons).

Wild-type C57BL/6 mice were crossed to generate the embryos used in this study.

## Lineage tracing, *in situ* hybridization, and cartilage staining

*Xenopus* embryos were fixed in MEMFA and stained with X-gal to visualize LacZ activity on the injected side before being processed for *in situ* hybridization. Whole mount *in situ* hybridization was performed on *Xenopus* embryos as previously described [68]. Synthetic digoxigenin-labeled antisense mRNAs probes were prepared using the T7 polymerase and pCS2-*fbl*, or pCS2-*ubtf*, or pCS2-*ncl*. NBT/BCIP was used as a color substrate. For *in situ* hybridization on mouse embryos, 20-µm cryostat sections of 4% paraformaldehyde-fixed, 30% sucrose/PBS-infused tissues frozen in gelatin (7.5% gelatin, 15% sucrose/PBS) were used. *In situ* hybridization experiments were performed as previously described [72]. The mouse FBL antisense probe was generated with the corresponding EST. The other antisense probes were generated with previously described templates encoding *krt12*, *sox2*, *msx1*, and *hairy2* [73], *twist* [74], *pax2*, and *pax6* [75], and *prdm12* [76]. Cartilage staining of *Xenopus* embryos was performed on tadpole heads as described [36].

## TUNEL assay

Terminal deoxynucleotidyl transferase dUTP nick end labeling (TUNEL) detects DNA breaks formed during apoptosis. Whole mount TUNEL staining of developmentally staged (neurula) control and morphant embryos was carried out as described in [77] with the following reagents: terminal deoxynucleotidyl transferase (TdT) (15 U/µl and 5x buffer, Thermofisher Ref. 10533065), digoxigenin-11-dUTP (25 nmol/25µl, Sigma Ref. 11093088910), Normal goat serum (Thermofisher Ref. 10000C), anti-dig alkaline phosphatase conjugate (Sigma Ref. 11093274910), nitro blue tetrazolium (Sigma Ref. 11383213001), and 5-bromo-4-chloro-3-indolyl phosphate (Sigma Ref. 10760994001). The control DMRT5 transcript was prepared as previously described [50].

## Western blot analysis

Total protein was extracted from single *Xenopus* embryos lysed with a mini-pestle (Vwr, Ref. 431–0094) in liquid nitrogen in a 1.5-ml Eppendorf tube and resuspended in ice-cold lysis buffer (50 mM Tris pH 7.6, 10 mM EDTA, 150 mM NaCl, 1% Triton, supplemented with complete protease inhibitor, Roche Ref. 11697498001–1 pellet per 50 ml buffer) and incubated on ice for 10 min. The lysate was cleared by centrifugation (12,000 rpm, 10 min at 4˚C). One embryo equivalent was loaded onto a 15% SDS-PAGE gel. The gels were transferred to

nitrocellulose membranes and probed in TBS/0.1% tween /3% BSA. Probing was as follows: anti-fibrillarin (Abcam, Ref. ab5821) incubated overnight at 1:2000 dilution, followed by anti-rabbit HRP (Cytiva, Ref. NA934V) used at a 1:5000 dilution for 2 hours; anti-Flag (Sigma, Ref. F3165) incubated overnight at 1:2000 dilution followed by anti-mouse HRP (Jackson ImmunoResearch, Ref. 115-035-062) used at 1:5000 for 2 hours; anti-alpha-tubulin (Sigma, Ref. T5168) incubated overnight at a 1:5000 dilution followed by anti-mouse HRP.

### RT-qPCR analysis

For RT-qPCR analysis of *X. laevis* animal caps, total RNA was extracted using the NETS method. The RNA samples were digested with RNase-free DNase I before RT- PCR. The amount of RNA isolated was quantified by measuring the optical density with a nanodrop spectrophotometer. The RNA samples were digested with RNase-free DNAse I before RT-PCR. cDNA was synthesized with the iScript cDNA synthesis kit (Biorad). Synthesized cDNA was amplified with the qScript One-Step SYBR Green qRT-PCR Kit (Quanta bio) with suitable primers (see S4 Table). Samples were normalized to *X. laevis gapdh*.

### Pre-rRNA analysis by northern blotting

Total RNA was extracted from MO-treated and untreated embryos with Tri reagent. Pooled stage 32 embryos were used. 5 μg total RNA was analyzed by northern blotting as previously described [46]. The probes used for the hybridization are depicted in S4 Table.

### Systematic mapping of ribosomal RNA 2'-O methylation by RiboMethSeq

RiboMethSeq analysis of rRNA 2'-O-methylation was performed as described in [60]. Briefly, ~100 ng of total RNA extracted from biological material was fragmented in bicarbonate buffer at pH 9.3 at 96˚C, fragmentation time was adapted to obtain short RNA fragments of ~20–30 nt in length. Fragmented RNA was end-repaired, to insure compatibility with adapter ligation. Library preparation was done using NEBNext Small RNA kit, according to manufacturer's recommendations. Barcoded libraries were loaded to the flow-cell and sequenced on Illumina HiSeq 1000. Raw reads were trimmed and aligned to the reference. Bioinformatics analysis was performed as described [60]. The complete data set is available at the European Nucleotide Archive under accession n˚PRJEB42253.

### Site-specific detection of 2'-O methylation by low dNTP primer extension

Primer extension was conducted as described in [78] on 5 μg total RNA extracted from stage 23 embryos. The primers used are listed in S4 Table.

### SnoRNA identification

To find snoRNAs in RNA-seq experiments, we downloaded sequencing data from lariat intronic RNAs of *X. laevis* [79]. The FASTQ file was downloaded from NCBI (SRR7615230, project PRJNA479418) with the SRA Toolkit package (http://ncbi.github.io/sra-tools/, version 2.1.10): *fastq-dump SRR7615230—outdir/path/to/dir*. FASTQ files were converted to FASTA files with seqtk software (https://github.com/lh3/seqtk, version 1.0–1) and used as our next input: *seqtk seq -a SRR7615230.fq.gz>SRR7615230.fasta*. We used snoScan (version 0.9.1) [80] to detect box C/D methylation guide snoRNAs in *X. laevis* RNA sequencing experiments. The predicted snoRNA sequence is provided in FASTA format. The snoRNA sequence consists of four leading nucleotides followed by the box C sequence, the sequence complementary to rRNA, the D box (and D' box) sequence, and two trailing nucleotides. snoScan computes a score based on

probabilistic models of sequence features (terminal stem, box C, guide sequence, box D, etc.). When multiple snoRNAs were predicted for one methylation site, we chose the one with the highest snoScan average bit scores and with high similarity to a corresponding known snoRNA in *X. tropicalis*. To validate our results, we checked if we detected the same snoRNAs as already identified by the Rfam database [81] in *X. laevis* (https://rfam.xfam.org/). We aligned snoScan snoRNA candidates with Rfam snoRNAs with Clustal Omega [82] (https://www.ebi.ac.uk/Tools/msa/clustalo/) and compared sequence alignment scores.

## Supporting information

**S1 Fig. RT-PCR analysis of *fbl*, *ubtf*, and *ncl* expression during *X. laevis* development.** Total RNA extracted from embryos at the indicated stages was analyzed by RT-PCR using amplicons specific to *fbl*, *ubtf*, or *ncl* transcripts (see Materials and Methods).
(TIF)

**S2 Fig. Relative retinal pigmented epithelium (RPE) area of data presented in Fig 2A.** Relative retinal pigmented epithelium (RPE) area (Kolmogorov-Smirnov test, **** = p<0.0001, ** = p<0.01).
(TIF)

**S3 Fig. Conservation of fibrillarin residues important for methyltransferase function.** The residue mutated in this work (D238 in *Xenopus laevis*) is highlighted in red in both panels. In the atomic resolution structure of *Archaeoglobus fulgidus* fibrillarin-Nop5 complex bound to its cofactor and methyl donor *S*-adenosyl-$_L$-methionine (AdoMet), it was shown that Asp-133 (equivalent to *Xenopus laevis* D238) is situated within 3.5 Å of the thiomethyl carbon of the bound AdoMet, implying that it plays a role as a catalytic residue [48]. When this residue was mutated to an alanine, the methylation activity of the complex was indeed totally abolished in an *in vitro* methylation assay [49]. It has been suggested that Asp-133 in fibrillarin may act as a general base by deprotonating the 2'-OH group of the target RNA during catalysis. It has further been suggested that Asp-133 may also facilitate cofactor binding through favorable electrostatic interactions [48,49]. **A**, 3-D model of the catalytic pocket of human fibrillarin (based on PDB 2ipx). D238 (in red, *Xenopus* numbering) is directly adjacent to the AdoMet (stick representation) with the methyl group to be transferred from the cofactor to the RNA substrate represented in pink. **B**, Multiple alignment between fibrillarin proteins of different origins (HUMAN, *Homo sapiens*; XENLA; *Xenopus laevis*; YEAST, *Saccharomyces cerevisiae*; ARCFU, *Archaeoglobus fulgidus*; PYRFU, *Pyrococcus furiosus*; and METJA, *Methanocaldococcus jannaschii*). Residues highlighted in blue and red (K/D/K) are absolutely conserved and correspond to the catalytic triad. The D residue in this triad is the residue mutated in this work. Bold, residues important for SAM binding. Asterisks, residues identical across all six species examined. Sequences were aligned with CLUSTAL.
(TIF)

**S4 Fig. *Xenopus* fibrillarin is required for small ribosomal subunit synthesis.** Total RNA extracted from stage 32 embryos injected into each cell at the 2-cell stage with *fbl* MO or with a non-targeting MO (Ctrl) was separated on denaturing agarose gel and processed for northern blotting with radioactively-labeled probes designed to detect pre-rRNA precursors. As a control, non-injected embryos were used. **A**, Ethidium-bromide-stained gel. Note that the mature 18S and 28S rRNAs appear as doublets, as previously described [83]. **B**, Northern blot analysis of pre-rRNA intermediates detected with probes specific to the 5'-ETS, the ITS1, and mature 18S rRNA (see panel C). **C**, Processing pathway in *Xenopus* [38]. Cleavage sites (A' to 5) are

indicated. The probes used in the northern blotting (panel B) are highlighted in color.
(TIF)

**S5 Fig. Clustering analysis illustrates the robustness of the RiboMethSeq data.** At each of the six developmental stages analyzed, four individual embryos were tested. Clustering of the RiboMethSeq analysis (here shown only for the 18S and 5.8S rRNA modifications; the same result was observed with 28S rRNA modifications) illustrates the remarkable robustness of our dataset. The staged embryos analyzed are depicted.
(TIF)

**S1 Table. List of all the 2'-O methylated positions identified in rRNA by RiboMethSeq in *X. laevis* during development.** The table provides the methylation score for each methylated position identified at each stage analyzed. Newly identified positions are highlighted in green. The new positions confirmed by low dNTP primer extension are highlighted in red. The positions modified in human rRNAs are indicated for reference. For simplicity, mean values for the four embryos analyzed at each stage are shown (for deconvoluted data see S2 Table).
(XLSX)

**S2 Table. Same as S1 Table, with deconvoluted data for each individual embryo analyzed at each stage.**
(XLSX)

**S3 Table. List of box C/D snoRNA candidates for the identified 2'-O methylated positions in *X. laevis*.** The sequence of box C/D snoRNAs assigned to each 2'-O methylated rRNA position is provided. SnoRNAs which were already present in the Rfam database are indicated by name. C box (blue), D box (green), D' box (purple), and complementary sequences (orange) are highlighted for each newly identified snoRNA. For reference, the positions modified in *X. tropicalis* and the identified snoRNAs in *X. tropicalis* are indicated (see [67] for details).
(XLSX)

**S4 Table. List of all the DNA oligonucleotides used in this work.**
(XLSX)

**S5 Table. Sequence of the morpholinos used in this work.**
(XLSX)

**S1 Data. Relative retinal pigmented epithelium (RPE) area data.** Primary data used to produce the graph in S2 Fig.
(XLSX)

## Acknowledgments

We thank Sadia Kricha (ULB) for assistance with *in vitro* transcript synthesis.

## Author Contributions

**Conceptualization:** Jonathan Delhermite, Lionel Tafforeau, Eric Bellefroid, Denis L. J. Lafontaine.

**Data curation:** Lionel Tafforeau, Denis L. J. Lafontaine.

**Formal analysis:** Lionel Tafforeau, Yuri Motorin, Eric Bellefroid, Denis L. J. Lafontaine.

**Funding acquisition:** Yuri Motorin, Eric Bellefroid, Denis L. J. Lafontaine.

**Investigation:** Jonathan Delhermite, Sunny Sharma, Virginie Marchand, Ludivine Wacheul, Ruben Lattuca, Simon Desiderio, Denis L. J. Lafontaine.

**Methodology:** Denis L. J. Lafontaine.

**Project administration:** Denis L. J. Lafontaine.

**Resources:** Denis L. J. Lafontaine.

**Supervision:** Lionel Tafforeau, Eric Bellefroid, Denis L. J. Lafontaine.

**Validation:** Lionel Tafforeau, Eric Bellefroid, Denis L. J. Lafontaine.

**Visualization:** Lionel Tafforeau, Yuri Motorin, Denis L. J. Lafontaine.

**Writing – original draft:** Lionel Tafforeau, Eric Bellefroid, Denis L. J. Lafontaine.

**Writing – review & editing:** Denis L. J. Lafontaine.

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
