## [Decision Letter · Decision Letter 0]

10 Feb 2021

Dear Dr Lafontaine,

Thank you very much for submitting your Research Article entitled 'Systematic mapping of rRNA 2’-O methylation during frog development and involvement of the methyltransferase fibrillarin in eye and craniofacial development in Xenopus laevis' to PLOS Genetics.

The manuscript was fully evaluated at the editorial level and by independent peer reviewers. The reviewers appreciated the attention to an important problem, but raised some substantial concerns about the current manuscript. Based on the reviews, we will not be able to accept this version of the manuscript, but we would be willing to review a much-revised version. We cannot, of course, promise publication at that time.

If you decide to revise the manuscript for further consideration at PLOS Genetics, please aim to resubmit within the next 60 days, unless it will take extra time to address the concerns of the reviewers, in which case we would appreciate an expected resubmission date by email to plosgenetics@plos.org.

[LINK]

We are sorry that we cannot be more positive about your manuscript at this stage. Please do not hesitate to contact us if you have any concerns or questions.

Yours sincerely,

Paul A Trainor

Guest Editor

PLOS Genetics

Gregory P. Copenhaver

Editor-in-Chief

PLOS Genetics

The reviewers raised significant concerns with the manuscript, which can possibly be addressed but will require considerable work. As reviewer 1 noted there is no mechanistic explanation for the cellular basis of the phenotype in frog fbl knockdown beyond arguing for a neural crest cell defect. Do the neural crest cells undergo apoptosis or is there a defect in proliferation and differentiation of neural crest cells. Determining where in the development process the neural crest cells are affected would go a long way toward accounting for the phenotype. Thus the disconnect between the phenotype and rDNA methylation was noted by reviewer 3 who also raised concerns about the data interpretation and recommended that conclusive proof could only come from demonstrating that rRNA is not methylated in an Fbl knockdown rescued with the Fbl-D238A mutant. Without this proof, the conclusion that rRNA through Fbl methylation is not needed for the development of NCC, but just Fbl in its role as ribosomal RNA chaperone, remains unconvincing. This would also help to address reviewer 2's comments who noted that the morphological defects were not apparently methylation dependent, but rather reflected a drastic loss of 18S rRNA. Thus the significance of any change in methylation remains unclear, and there needs to be a better connection between the two distinct components of the work.

Reviewer's Responses to Questions

**Comments to the Authors:**

Reviewer #1: Ribosomopathies are rare congenital disorders that are caused due to impaired ribosome biogenesis and in most cases affect specific tissue types. The underlying cause of ribosomopathies could be due to one of the two genetic mechanisms (1) mutations in ribosomal proteins (2) mutations in ribosome biogenesis factors. Our understanding of the factors essential for ribosome biogenesis, especially proteins required for transcription and processing of ribosomal RNAs (rRNA) remains incomplete, hence further study is needed. This manuscript explores the requirement of Fibrillarin, a rRNA processing factor, during xenopus embryonic development. The authors observe craniofacial and ocular defects in Fibrillarin morpholino knockdown embryos and identify that neural crest migration is affected due to Fibrillarin knockdown. In addition, 18S rRNA processing is impaired in the knockdown embryos, indicating that Fibrillarin is essential for rRNA processing in xenopus. Given the nature of the phenotype in fish and mouse mutants of Fibrillarin and now frog knockdown, it indicates that the function of Fibrillarin is conserved in vertebrates. However, in this manuscript there is no explanation for the cellular basis of the mechanism of the phenotype in frog fbl knockdown. Do the neural crest cells undergo apoptosis or is there a defect in proliferation and differentiation of neural crest cells?

In the second part of the paper, the authors determine previously identified and novel 2' O-Methylation sites in rRNA and identify new snoRNA. However, how this connects to the fbl mutant and the phenotype of the fbl mutant is unclear. Does the absence of Fibrillarin change specific methylation of rRNA and are those methylation marks craniofacial specific?

Other minor suggestions:

1. Legend for Figure 2 mentions that embryos were injected with morpholinos at one cell stage. Should it be two-four cell stage instead?

2. The figure quality for 2B and 2C could be improved.

3. Figure 1B: It might be easier to interpret the results if the bars were grouped based on gene instead of the condition for the animal cap explant.

Reviewer #2: The authors set out to determine the molecular basis of inherited ribosomopathies and the potential importance of differential rRNA 2’-O-methylation during development, investigating this through depletion of the Fbl methyltransferase. The authors initially report the detection of transcripts encoding a number of ribosome synthesis factors, including FBL in embryos, with uneven distribution and enrichment in NCC cells.

FBL mRNA depletion dramatically impaired 18S rRNA maturation in Xenopus embryos, consistent with previous findings in other systems. This was associated with striking morphological defects, presumably due to altered or reduced translation. Unexpectedly, these changes were largely restored by expression of a methylation defective FLB protein. The authors use RiboMethSeq to reanalyze rRNA methylation in detail during development. This identified 15 putative novel methylation sites, 10 of which were confirmed by primer extension. Notably several were apparently differentially modified during development.

The work is technically very good, the MS is well written, and the data on methylation will be of value. The obvious weakness in the overall structure of the MS, is that the morphological defects are apparently not methylation dependent, but rather reflect a drastic loss of 18S rRNA. In consequence, the changes in methylation are of unclear significance.

Overall, the MS presents interesting and useful data on the spatial distribution of ribosome synthesis factors, morphological effects of reduced ribosome synthesis during development and changes in modification. Better linking these two sections would significantly strengthen the MS.

Specific comments:

1. It was unexpected that the morphological changes were largely restored by expression of a methylation defective FLB protein. This indicates that residual FBL expression on knockdown is sufficient to support some, presumably adequate, level of methylation that supports ribosome function. It would be useful to indicate what, if any, defects are seen in cells complemented by expression of the catalytic mutant, since these are likely due to a lack of methylation. Their description and discussion would potentially link the tow section of the paper. It would also be helpful to indicate the residual level of wt FBL in the complemented animals. Ideally, RiboMethSeq might indicate what level of methylation is sufficient to support close to normal development. However, this may not be feasible at present.

2: Figs. 1, S1 and associated text: The text has several sections with statements like ”…suggesting that Fbl plays an important role at these locations”. As FBL is believed to be essential for ribosome synthesis in all cells, this statement seems inappropriate. It would be better to make the interpretation more explicit. Presumably what the authors intend to conclude is something like: elevated FBL transcript abundance may be associated with increased FBL protein synthesis. If FBL is limiting for ribosome synthesis, this increased production might alter the rate or pathway of pre-rRNA packaging, processing or modification. This point is raised in the Discussion, but it would be better to be explicit from the start.

Reviewer #3: This article consists of two, somehow distinct, messages. First, the expression of the protein fibrillarin is quantified at different stages during the development of Xenopus laevis. Fbl knockdown is found to cause morphological defects, which are attributed to defects in the development of neural crest cells. Second, the authors quantify site-specific methylations of ribosomal RNA during development, identify additional fifteen positions and observe a (slight) increase in hypomethylation in the late stages of development. The manuscript provides some new data relative to Xenopus laevis, which are found to be consistent with the literature reports on other species. The level of novelty is limited. Nevertheless the data are worth publishing.

All in all, I have doubts on the demonstration that the defects in NCC observed upon Fbl knockdown are not due to the Fbl methylation activity. First, the rescue effect with the methylation incompetent mutant is far from being complete; second, this mutant has been demonstrated to be methylation incompetent in archaea fibrillarin but not in eukaryotic fibrillarin. Even if the sequence conservation of fibrillarin is high across species, the sequence conservation of the N-terminal domain of the Nop proteins is not; this may cause subtle rearrangements in the enzyme that change the mutant activity relationships. A much more convincing proof would be demonstrating that the rRNA is indeed not methylated in the Fbl knockdown rescued with the Fbl-D238A mutant. Without this proof, the conclusion that rRNA through Fbl methylation is not needed for the development of NCC, but just Fbl in its role as ribosomal RNA chaperone, remains unconvincing.

**Have all data underlying the figures and results presented in the manuscript been provided?**

Reviewer #1: Yes

Reviewer #2: Yes

Reviewer #3: Yes

PLOS authors have the option to publish the peer review history of their article (what does this mean?). If published, this will include your full peer review and any attached files.

Reviewer #1: No

Reviewer #2: **Yes: **David Tollervey

Reviewer #3: No

---

## [Decision Letter · Decision Letter 1]

28 Sep 2021

Dear Dr Lafontaine,

Thank you very much for submitting your Research Article entitled 'Systematic mapping of rRNA 2’-O methylation during frog development and involvement of the methyltransferase fibrillarin in eye and craniofacial development in Xenopus laevis' to PLOS Genetics.

The manuscript was fully evaluated at the editorial level and by independent peer reviewers. The reviewers appreciated the attention to an important problem, but raised some substantial concerns about the current manuscript. Based on the reviews, we will not be able to accept this version of the manuscript, but we would be willing to review a much-revised version. We cannot, of course, promise publication at that time.

If you decide to revise the manuscript for further consideration at PLOS Genetics, please aim to resubmit within the next 60 days, unless it will take extra time to address the concerns of the reviewers, in which case we would appreciate an expected resubmission date by email to plosgenetics@plos.org.

[LINK]

We are sorry that we cannot be more positive about your manuscript at this stage. Please do not hesitate to contact us if you have any concerns or questions.

Yours sincerely,

Paul A Trainor

Guest Editor

PLOS Genetics

Gregory P. Copenhaver

Editor-in-Chief

PLOS Genetics

While the reviewers recognize the improvements you have made to the original manuscript there are two key issues that they note as remaining outstanding. The first is that the biological insights of the manuscript remain somewhat limited. The second is the lack on integration between the phenotypic data resulting from fibrillarin MO knockdown and the absence of an effect on rDNA methlyation. There is also an absence of co-localization of cell death and other parameters with neural crest cells as is commonly presented in zebrafish or mouse models of ribosomopathies. I recognize that the authors have toned down their statement that fibrillarin is required for rDNA methylation or that defects in rDNA methylation underpin the phenotype. However, the fact that the fibrillarin morphant phenotype can be rescued with a methylase dead fibrillarin construct suggests that the fibrillarin morphant phenotype is not caused by defects rDNA methylation. The flow of results in the papers moves from phenotype to methylation to phenotype to preRNA processing, to ribosome biogenesis and p53 rescue which makes the paper difficult to follow. It might make more sense for the methylase dead fibrillarin rescue to come after the evaluation of rDNA methylation sites such the rescue would allow the authors to conclude the phenotype is not due to deficient methylation, but rather deficient ribosome biogenesis. The authors could further demonstrate this by testing what happens to the rRNA species post fibrillarin rescue. Is processing and are the levels return to normal. This might help explain why only a 10-20% increase in fibrillarin levels from the 30% baseline of the morphants is sufficient to elicit a substantial rescue.

Reviewer's Responses to Questions

**Comments to the Authors:**

Reviewer #1: While the manuscript has definitely improved from the first submission, here are two points of concern:

1. Since the cellular basis for the phenotype of fbl mutant is apoptosis, the defect is lack of NCC survival rather than maturation.

2. In addition, since there is no direct connection between fbl and 2’ O-Methylation data, the paper seems like two different stories. One about Fibrillarin’s requirement in craniofacial development and the other about rRNA modification. I am not convinced that the two data belong together.

Reviewer #2: The MS has been improved and my specific comments have been suitably addressed. I am happy to support publication of the revised MS.

**Have all data underlying the figures and results presented in the manuscript been provided?**

Reviewer #1: Yes

Reviewer #2: Yes

PLOS authors have the option to publish the peer review history of their article (what does this mean?). If published, this will include your full peer review and any attached files.

Reviewer #1: No

Reviewer #2: No

---

## [Decision Letter · Decision Letter 2]

23 Dec 2021

Dear Dr Lafontaine,

We are pleased to inform you that your manuscript entitled "Systematic mapping of rRNA 2’-O methylation during frog development and involvement of the methyltransferase Fibrillarin in eye and craniofacial development in Xenopus laevis" has been editorially accepted for publication in PLOS Genetics. Congratulations!

Yours sincerely,

Paul A Trainor

Guest Editor

PLOS Genetics

Gregory P. Copenhaver

Editor-in-Chief

PLOS Genetics

Comments from the reviewers (if applicable):

Reviewer's Responses to Questions

**Comments to the Authors:**

Reviewer #1: The authors and I seem to disagree on whether the story in the current manuscript is cohesive. I agree that given the known roles of Fibrillarin, it is important to assess the status of methylation on rRNA. However, since the methylation status was identified in WT animals and not in Fibrillarin morphants, the function of Fibrillarin in this particular context is unknown. If we assess the morphants, will we identify that only 10 out of 103 methylation sites are Fibrillarin dependent or maybe all of them are Fibrillarin dependent? Without that information, the biological context of the methylation sites is missing and doesn’t necessarily relate to the initial part of the paper which describes how Fibrillarin regulates craniofacial development. Although from the author response I understand that this experiment is out of the scope of this paper, the authors have the required animal model and the experimental setup to answer this particular question for a future publication perhaps.

The data presented in the current paper is of good quality without any fundamental flaws. Therefore, even though the author and I have a difference in opinions of what the story should be, the paper should be published.

**Have all data underlying the figures and results presented in the manuscript been provided?**

Reviewer #1: Yes

PLOS authors have the option to publish the peer review history of their article (what does this mean?). If published, this will include your full peer review and any attached files.

Reviewer #1: No

**Data Deposition**

http://datadryad.org/submit?journalID=pgenetics&manu=PGENETICS-D-20-01958R2

**Press Queries**

---

## [Editor Report · Acceptance letter]

14 Jan 2022

PGENETICS-D-20-01958R2 

Systematic mapping of rRNA 2’-O methylation during frog development and involvement of the methyltransferase Fibrillarin in eye and craniofacial development in </i>Xenopus laevis</i> 

Dear Dr Lafontaine, 

We are pleased to inform you that your manuscript entitled "Systematic mapping of rRNA 2’-O methylation during frog development and involvement of the methyltransferase Fibrillarin in eye and craniofacial development in </i>Xenopus laevis</i>" has been formally accepted for publication in PLOS Genetics! Your manuscript is now with our production department and you will be notified of the publication date in due course.

With kind regards,

Zsofia Freund

PLOS Genetics

On behalf of:
